# Effect of Sulfur Modification on Structural and Electrochemical Performance of Pitch-Based Carbon Materials for Lithium/Sodium Ion Batteries

**DOI:** 10.3390/nano14171410

**Published:** 2024-08-28

**Authors:** Zihui Ma, Zhe Wen, Yan Song, Tao Yang, Xiaodong Tian, Jinru Wu, Yaxiong Liu, Zhanjun Liu, Huiqi Wang

**Affiliations:** 1CAS Key Laboratory of Carbon Materials, Institute of Coal Chemistry, Chinese Academy of Sciences, Taiyuan 030001, China; mazihui21@mails.ucas.ac.cn (Z.M.); wenzhecn@126.com (Z.W.); taomung@126.com (T.Y.); tianxiaodong0124@163.com (X.T.); jinruwu8801@163.com (J.W.); zjliu03@sxicc.ac.cn (Z.L.); 2College of Energy and Power Engineering, North University of China, Taiyuan 030051, China; 3Center of Materials Science and Optoelectronics Engineering, University of the Chinese Academy of Sciences, Beijing 100049, China

**Keywords:** coal tar pitch, sulfurized pitch, lithium-ion batteries, sodium-ion batteries

## Abstract

Coal tar pitch (CTP) has become an ideal choice in the preparation of anode precursors for lithium-ion batteries (LIBs) and sodium-ion batteries (SIBs) because of its abundant carbon content, competitive pricing and adjustable structure properties. In this paper, sulfurized pitch-based carbon (SPC-800) was obtained by allowing CTP to react with sulfur at 350 °C and subsequently achieve carbonization at 800 °C. SPC-800 was more disordered and had a larger layer spacing than carbonized CTP (PC-800). Upon utilization as an anode for LIBs, SPC-800 possessed a higher reversible specific capacity (478.1 mAh g^−1^ at 0.1 A g^−1^), while utilization in SIBs displayed a capacity of 220.9 mAh g^−1^ at 20 mA g^−1^. This work is an important guide to the design of high-performance anodes suitable for use with both LIBs and SIBs.

## 1. Introduction

The growing global demand for renewable energy sources has resulted in chemical energy storage technologies becoming essential [1]. Lithium-ion batteries (LIBs) and sodium-ion batteries (SIBs) are anticipated as energy storage devices with immense growth potential [2]. LIBs and SIBs work on similar principles, with their anode material as a pivotal factor in their performance [3,4]. Hard carbon materials are of particular interest, owing to their large layer spacing and inherent porous- and defect-rich structure [5].

Coal tar pitch (CTP) is the principal by-product resulting from the graded distillation process, and it primarily consists of polycyclic aromatics with different molecular weights [6]. Due to its abundant availability, cost-effectiveness, and significant residual carbon content, CTP is considered a promising candidate for carbon precursors [7]. However, during the carbonization process, the aromatic nuclei and the side chains undergo condensation and rearrangement to form nearly parallel stacked carbon sheets, which causes a tendency to graphitization under conditions of elevated temperatures [8]. The high degree of graphitization with small layer spacing usually leads to lowered specific capacity that cannot meet the requirements of energy storage devices. As a result, the manipulation of pitch microstructures has become a widely adopted approach for preparing carbon materials with a superior performance. Hu et al. found a structural transformation from order to disorder by a simple pre-oxidation treatment of pitch [9]. In this process, oxygen-functional groups play a crucial role. At low temperatures, they promote the cross-linking reaction, while as the temperature rises to the carbonization stage, they maintain carbon structure’s stability and prevent structural changes in carbon structures. Daher et al. also pointed out that the graphitization of pitch during pyrolysis can be effectively inhibited by pre-oxidation treatment, resulting in an amorphous microstructure [10].

Due to being in the same main group, sulfur has many similar properties to oxygen. In practical applications, sulfur is more accessible to operate and control than oxygen. Kong et al. successfully prepared sulfur-doped graphene material using sulfur-containing polyacetylene monomers, which displayed excellent performance as the anodes for LIBs [11]. Zou et al. successfully prepared sulfur-doped nanocarbon sheets with 0.41 nm layer spacing by employing sodium dodecyl sulfate [12]. After undergoing 100 cycles of charging and discharging at 0.1 A g^−1^, the material attained a capacity of 321.8 mAh g^−1^. The presence of sulfur effectively increased the interlayer’s ability to speed up the ion insertion and extraction process, thus enhancing its electrical conductivity. Furthermore, the presence of sulfur can serve as an additional absorption site for carbon materials, which enhances the specific capacity of the materials and thus their energy storage capacity [13,14]. Yin et al. successfully synthesized sulfur-doped carbon materials utilizing pitch that sustained its specific capacity, remaining at 120 mAh g^−1^ even through 2000 cycles at 2 A g^−1^ in SIBs [15]. He et al. successfully manufactured sulfur-doped pitch-based carbon via thermal treatment of pitch with sulfur. The material exhibited a specific capacity reaching 482.8 m Ah g^−1^ at 0.1 A g^−1^ in SIBs [16]. However, their studies mainly explored the electrochemical properties and applications of pitch-based carbon from the viewpoint of sulfur doping. Comprehensive and profound investigations into the interaction between sulfur and pitch, the structural transformations triggered by sulfur during carbonization, and their corresponding effects on electrochemical properties are still lacking.

In this work, a straightforward and economical approach was adopted for the manufacturing process of sulfurized pitch-based hard carbon. The effects of sulfur modification on the composition, microstructure, and lithium/sodium storage properties were studied. The prepared carbon material demonstrated a superior capacity for LIBs and SIBs that was superior to un-modified material. Our work may provide scientific guidance for designing carbon anodes for LIBs and SIBs.

## 2. Experimental

### 2.1. Materials Preparation

CTP (Shanxi Yongdong Chemical Company (Yuncheng, China); Characterization parameters, including elemental analysis and softening point (SP), are outlined in Table 1) was employed as the raw material. Sublimed sulfur (Sinopharm Chemical Reagent Limited corporation, Shanghai, China) was utilized as the cross-linker. The preparation of sulfurized pitch carbon is depicted via a graph in Figure 1. Mixtures of 4 g, 8 g and 12 g of sublimed sulfur and 20 g of CTP were each heated to 350 °C for 3 h under an argon atmosphere for vulcanization. The obtained samples were named SPC-4, SPC-8 and SPC-12, respectively. Afterwards, the chosen sulfurized pitch carbon (SPC-800) was synthesized through the carbonization of SPC-8 at 800 °C in an argon atmosphere for 2 h (SPC-800). For comparison, pitch carbon was prepared by treatment of CTP at 350 °C without the addition of sulfur. The corresponding carbonized product was labeled as PC-800.

### 2.2. Materials Characterization

The elemental analyses (EA) were conducted using a Vario EL CUBE. The morphology of the samples was analyzed by a JMS-7001F scanning electron microscope (SEM, Tokyo, Japan) and a JEM-2001F high-resolution transmission electron microscope (HRTEM). X-ray photoelectron spectroscopies (XPS, AXIS ULTRA DLD with monochromatic Al Kα as the incident X-rays) was collected to identify surface elements and functional groups. Thermogravimetry analysis (TGA) were performed under Ar at a heating rate of 10 °C min^−1^ inside a HITACHI STA200 (Netzsch, Selb, Germany). The functional groups’ composition were determined via Fourier transform infrared spectroscopy (FT-IR, Bruker tensor 27, Berlin, Germany). The structural characteristics were found through X-ray diffraction (XRD, AXS D8 ADVANCE A25, Berlin, Germany, with Cu Kα radiation sources) and Raman spectroscopy (HORIBA Lab RAMHR Evolution, Kyoto, Japan).

### 2.3. Electrochemical Characterization

The electrochemical performance measurement was obtained using LAND CT2001A battry test system. A CHI 660E working station was utilized for Cyclic voltammetry (CV) measurements.

For the preparation of LIBs, the carbon material (active substance), Polyvinylidene fluoride (PVDF) (binder) and carbon black (conductive agent) were mixed in an 8:1:1 ratio, dissolved in N-methyl Pyrrolidone (NMP) to create a slurry, and then coated onto copper foil to manufacture working electrodes. Within an argon-filled glove box, CR2016 button batteries were assembled using LiPF_6_ electrolytes, Li foil and Celgard 2400 membranes as the counter-electrodes and diaphragm, respectively.

For the SIBs, the carbon material (active substance), carbon black (conductive agent), sodium carboxymethyl cellulose (CMC) and styrene butadiene rubber emulsion (SBR) (binder) were blended in a 91:2:2:5 ratio, dispersed in deionized water, and then coated on copper foil. CR2032 button batteries were installed in a glove box utilizing NaPF_6_ electrolytes, sodium foil and glass fibers as the counter-electrodes and diaphragm, respectively.

## 3. Results and Discussion

### 3.1. Structural Characterization

Different doping amounts of sublimated sulfur had different cross-linking effects on pitch. Appendix A shows the thermogravimetric curves of SPC-4, SPC-8 and SPC-12. The rise in sublimation sulfur doping led to an initial growth followed by a decline in the carbon residue rate. Appendix A demonstrates a trend where the C/H ratio initially rose and then fell as the S element concentration increased. During one of the instances when the quantity of sulfur sublimated reached 12 g, the percentage of S element in the system did not experience a sudden rise, possibly due to an excess of S element not engaging in the system’s cross-linking reaction. Simultaneously, an excessive amount of sulfur atoms impeded the cross-link process among pitch molecules, resulting in a reduction in the polymerization degree of pitch products. Following an investigation, it was determined that SPC-8 exhibited superior thermal stability and a relatively high degree of polymerization. Consequently, SPC-8 was selected as the precursor for the ensuing carbonization process, and its name was shortened to SPC.

Figure 2a shows the schematic diagram of the synthesis of SPC. To illustrate the function of sulfur atoms in pitch cross-linking visually, we chose the simplest single S atom cross-linking for the schematic diagram. In fact, single S atom cross-linking is the most ideal method because the reaction system is more likely to form two or more sulfur chain structures. The thermogravimetric (TG) profiles are shown in Figure 2b. It was found that both samples exhibited one-stage weight loss. After the heating treatment, the residue rate of SPC was maintained up to 88.13% (at 800 °C, while that of PC was 39.84% at 535 °C), demonstrating that the residue of SPC was higher than PC. The addition of sulfur promoted the cross-linking of small molecules and thus contributed to high thermal stability [9]. The EA results (Table 2) show that the mass fraction of sulfur in SPC was as high as 15.3%. The XRD patterns (Figure 2c) reveal that no peak of sulfur appeared for SPC, which indicated the cross-link of sulfur and pitch molecules [16]. For the FT-IR spectra of PC and SPC (Figure 2d), symmetrical and asymmetric stretching bands of −CH_2_−at around 2860 and 2920 cm^−1^ were noticed. The bands situated at 1035 cm^−1^ and 1160 cm^−1^ in SPC were significantly enhanced, which could be attributed to the tensile vibration of the S−O and C−S bonds [17,18]. By contrast, the aromatic C−H bond located at 1380 cm^−1^ in SPC was weaker than the corresponding bond in PC. The C/H ratio, an essential parameter for the characterization of pitch aromatic structures, increased from 1.89 for PC to 2.57 for SPC (Table 2). These changes highlight the aliphatic hydrocarbon fracture and the increasing polycondensation degree of aromatic rings after the introduction of sulfur [19]. The results proved the dehydrogenation condensation in the vulcanizing process of pitch. The addition of sulfur acted as a free radical initiator, thus promoting the condensation of aromatic molecules.

The XPS spectra, shown in Figure 2e, were employed to determine the elemental surface composition and functional group evolution of PC and SPC. It was found that the survey spectrum of SPC contained carbon, oxygen and sulfur atoms, whereas only C and O atoms appeared in PC, indicating a successful addition of S atoms to the SPC. Peaks of S 2s (228 eV) and S 2p (164 eV) were observed for SPC [15]. The high-resolution C 1s spectrum of SPC, as shown in Figure 2f, revealed four peaks at 284.6, 285.2, 286.5 and 288.3 eV, which were attributed to the surface-active functional groups of C−C/C=C, C−S, C−O, and C−C=O, respectively. In contrast to PC, the C−S group at binding energy (B.E.) of 285.2 eV was observed for SPC (Figure 2f) [20]. The high-resolution S 2p spectrum for SPC was conducted to identify the chemical state of sulfur. It could be fitted into six distinct peaks, with binding energies of 161.8 and 163.1, 163.6 and 164.5, and 166.1 and 167.7, which corresponded to the S2p_3/2_ and S2p_1/2_ of the C=S group, sulfur (C−S_X_−C, x = 1, 2) covalent bond and S-oxide (C−SO_X_−C, x = 2, 4), respectively [21,22]. The results indicated that sulfur exists in the form of sulfur-containing bonds in pitch, which aligns well with the results obtained from FT-IR and XRD analysis.

After carefully considering the results, it can be conclusively stated that the introduction of sulfur into CTP had two effects. One effect was to act as a free radical initiator that promoted free radical formation by removing aromatic hydrogen and making polymerization easier. The second was that sulfur was directly involved in polymerization to form sulfur-containing bonds, thus causing the merging of small molecules into large molecules. Therefore, the reaction mechanism between sulfur and pitch is as follows:(1)$$ArH+S\to Ar-Ar+H_{2}S$$
(2)$$ArH+S\to Ar-S_{X}-Ar+H_{2}S$$
where $S_{X}$ represents the sulfur (C−S_X_−C, x = 1, 2) and ArH represents the aromatic compound.

Figure 3a,b show the morphology of PC and SPC. It was found that both PC and SPC were shaped like irregular chunks with small particles attached. The uniform distribution of carbon and sulfur throughout the samples indicates the successful introduction of sulfur into the pitch (Figure 3c,d). It is observed from Figure 3e,f that SPC-800 maintained its original morphology, while numerous irregular small pieces appeared for PC-800. This is attributed to the cross-linked structure generated by sulfur and pitch reactions that endowed the sample with a more stable structure. According to the HRTEM images (Figure 3g,h), PC-800 possessed parallel carbon layers exhibiting some long-range orders. Additionally, the selected area electron diffraction (SAED) map showed clear diffraction rings. However, SPC-800 demonstrated short-range ordered structures with long-range disorder, and the diffraction rings were blurred. These results suggest that sulfur modification adversely affected the graphitization process of the pitch.

The XPS spectra of both PC-800 and SPC-800 are illustrated in Figure 4a. Figure 4b shows the S2p fitting results for SPC-800; it can be observed that S atoms exist primarily in the forms of C−SO_x_−C and C−S_x_−C. As shown in Appendix A, a high-resolution C 1s spectrum of SPC-800 revealed four peaks at 284.5, 285.2, 286.6 and 288.4 eV, which were attributed to the surface-active functional groups of C−C/C=C, C−S, C−O, and C−C=O, respectively. In contrast to PC-800, the C−S group at a binding energy (B.E.) of 285.2 eV was observed for SPC-800. S atoms were covalently connected in the carbon framework. Appendix A indicates the element content proportion of PC-800 and SPC-800. It is undeniable that the O content of SPC-800 is slightly higher than that of PC-800, which may be due to the reaction between S and some O elements during the heat treatment, causing a certain amount of O to be retained in the system. Introducing the S atom with its larger atomic radius effectively increased the carbon layer spacing [23]. The peaks observed around 23° and 43° might be attributed to the (002) and (100) planes of amorphous carbon for the XRD patterns (Figure 4c) [24]. By comparing with the XRD data of standard graphite samples, we can calculate the layer spacing (*d*_002_), the average width of aromatic sheets ($L_{a}$), and the average stacking height of aromatic sheets ($L_{c}$). It is notable that SPC-800 displayed broader peaks around 23° and 43°, indicating a more disordered structure compared to PC-800 [25]. The layer spacing of SPC-800 was about 0.358 nm (0.353 nm for PC-800). Appendix A shows the lamellar structure parameters of PC-800 and SPC-800. They indicate that the addition of sulfur hindered the orderly growth of pitch molecules at high temperatures. The Raman patterns (Figure 4d) exhibit two distinctive peaks around 1350 cm^−1^ and 1590 cm^−1^. Based on the Lorentz function, the Raman spectra from 950–1750 cm^−1^ could be back-convoluted into four subpeaks located at 1220, 1350, 1500, and 1590 cm^−1^, representing D_4_, D, D_3_, and G bands. The D_4_ band was associated with C−C and C=C vibrations of the sp^2^−sp^3^ bond or polyene structure, encompassing non-isotropic carbon structures arising from doping. The D band was assigned to carbon atoms situated at the edges and defects of graphene layers. The D_3_ band could be ascribed to vibrations of sp3 hybridized amorphous carbon atoms, while the G band might result from E_2g_ lattice stretching vibrations of ideal graphite [26,27]. I_D_/I_G_ has been commonly applied as an indicator to assess the extent of defects or disorders in carbon-based materials. The I_D_/I_G_ ratio of SPC-800 (1.97) was higher than that of PC-800 (1.85), which indicates that more defects were present in the SPC-800 sample than in the PC-800 sample. It corresponded to the results obtained from the XRD and TEM analyses. For SPC-800, the introduction of sulfur interfered with the otherwise orderly arrangement of the carbon layers, ultimately leading to the generation of a disordered carbon structure.

### 3.2. Electrochemical Performance

#### 3.2.1. Electrochemical Performance of LIBs

Figure 5 shows the electrochemical performance of PC-800 and SPC-800 in Li^+^ half-cells. The cyclic voltammetry (CV) curve of SPC-800 (Figure 5a) shows two distinct cathodic peaks in the initial cycle specifically located at about 0.01 V and 0.75 V. The presence of a peak of approximately 0.01 V indicates the insertion of Li^+^ into the carbon substrate [28]. The peak observed near 0.75 V could be indicative of the generation of a solid electrolyte interface (SEI) film that disappeared during subsequent cycling [29]. In addition, there was an oxidation peak that emerged around 1.5 V during the first cycle that shifted to the right in the subsequent cycle. This phenomenon could be ascribed to the reversible interaction between Li^+^ and sulfur [30]. Figure 5b exhibits the charge and discharge curves for both PC-800 and SPC-800. At 0.1 A g^−1^, SPC-800 demonstrated an impressive reversible capacity of 478.1 mAh g^−1^ when compared to PC-800’s lower capacity of 270.6 mAh g^−1^. Moreover, SPC-800 exhibited Li^+^ storage capacities of 451.8, 351.3, 281.4, 240.1, 204.3 and 150.5 mAh g^−1^ at 0.1, 0.2, 0.5, 1, 2 and 5 A g^−1^, respectively. Subsequently, when the current density was dialed back to 0.1 A g^−1^, SPC-800 regained a specific capacity of 454.3 mAh g^−1^. The SPC-800 anode exhibited an excellent performance rate in comparison to other known carbon anodes (Table 3 and Appendix A), suggesting good prospects for its future application [9,31,32,33,34,35]. At 2 A g^−1^, SPC-800 demonstrated an outstanding capacity retention rate (almost 100%) throughout the initial 100 cycles, and after 500 cycles ultimately achieved a capacity of 225.1 mAh g^−1^. This performance fully verifies that SPC-800 possessed fast ionic kinetics and good cyclic stability (Figure 5c,d).

The CV curves of SPC-800 at scan rates 0.1–1.0 mV s^−1^ are shown in Figure 6a. The CV curve shape remained almost constant as the scan rate gradually increased, which significantly demonstrates that SPC-800 exhibited excellent stability [36]. The power-law equation ($i=av^{b}$) is an essential instrument for analysing lithium storage mechanisms. In this equation, a and b were correlated primarily with the intercept and slope of the log(i)−log(v) profile, while *i* was the peak current and *v* was the scan rate [37]. The variable *b* acts as a critical factor for evaluating an electrode’s energy storage mechanism. Specifically, if the b value is near 1, this signifies that the capacity of the electrode mainly originates from the surface-induced capacitive process (SCP). When the b value is closer to 0.5, this suggests that the diffusion-controlled intercalation process (DIP) is the main contributor [38,39]. After calculation, the *b* values of PC-800 and SPC-800 were 0.80096 and 0.64127, respectively, indicating that SCP and DIP jointly controlled the energy storage of both PC-800 and SPC-800. Notably, the *b* value of SPC-800 was lower than that of PC-800, suggesting a more significant DIP in SPC-800. The following formula could further quantify DIP behavior [40,41].
(3)$$i_{v}=k_{1}v+k_{2}v^{\frac{1}{2}}$$
where $k_{1}$ and $k_{2}$ are constants, *k*_1_*v* denotes the contribution of capacitive control to the overall capacity, and $k_{2}v^{\frac{1}{2}}$ indicates the diffusion-based contribution. Figure 6d exhibits the percentage contribution of capacitive behavior to the overall capacity across varying scan rates. In particular, the SCP contributions of PC-800 and SPC-800 were significantly enhanced with an increasing scan rate. It is worth mentioning that PC-800 demonstrated a higher SCP than SPC-800’s contribution at each scan rate. To be precise, the capacitive contribution of PC-800 increased significantly from 60% to 81% as the scan rate was increased from 0.1 mV s^−1^ to 1.0 mV s^−1^. In contrast, SPC-800’s capacitive contribution only increased from 41% to 69%.

#### 3.2.2. Electrochemical Performance of SIBs

SPC-800 is also suitable for application as an anode in SIBs. The CV curves for both PC-800 and SPC-800 are recorded in Figure 7a,d. Both curves exhibit a characteristic oxidation peak near 2.0 V accompanied by a reduction peak in the vicinity of 1.0 V. These phenomena were attributed to the redox reactions during the initial stages of charging and discharging [14]. Both curves show an oxidation peak situated around 2.2 V, which corresponded with CuO formation [42]. The source of the copper was attributed to the current collector made of copper foil [43]. In addition, both samples exhibited a distinct irreversible peak near 0.2 V, which vanished in subsequent cycles. This observation was primarily attributed to SEI film formation. Figure 7b,e display the first three charge-discharge curves of PC-800 and SPC-800 under 20 mA g^−1^. In the initial discharge phase, PC-800 reached a capacity of 134.8 mAh g^−1^, whereas SPC-800 surpassed this with 286.5 mAh g^−1^. According to Figure 7c, SPC-800 exhibited reversible specific capacities of 207.8, 183.2, 171.1, 157.4, 135.7, 112.9 and 88.7 mAh g^−1^ under 0.02, 0.05, 0.1, 0.2, 1 and 2 A g^−1^, respectively. Notably, the reversible specific capacity of SPC-800 turned to 190 mAh g^−1^ as the current density reversed to 0.02 A g^−1^, demonstrating remarkable electrochemical reversibility and robustness. In addition, SPC-800 demonstrated exceptional cycling durability, retaining a capacity of 102.1 mAh g^−1^ even after completing 300 cycles conducted at 1 A g^−1^. The kinetic perspectives of both PC-800 and SPC-800 were explored to gain insights into their sodium storage mechanisms. The *b* values of PC-800 and SPC-800 were 0.75105 and 0.58137, respectively (Figure 7g). Figure 7h shows the contribution of the capacitive behavior to the capacity of SPC-800 at a 1 mV s^−1^ scan rate. The relative proportion of the capacitive behavior to capacity across various scan rates is shown in Figure 7i. Interestingly, although the SCP contributions of PC-800 and SPC-800 increased with the scan rate, it was observed that the SCP contribution of PC-800 was higher than that of SPC-800 across various scan rates. The capacitance contribution of the PC-800 anodes rose from 72% to 91% by changing the scan rate from 0.1 mV s^−1^ to 1.0 mV s^−1^. In contrast, that of SPC-800 only increased from 46% to 68%. This result again demonstrates that the sulfurized pitch improved the Na^+^ diffusion process, aligning with the analytical results presented in Figure 7g. The conclusion was also in accordance with the findings in LIBs.

### 3.3. Relationship between Structure and Electrochemical Performance

Drawing from the preceding analysis, the superior electrochemical performance exhibited by sulfur-modified pitch-based carbon (SPC-800) is primarily accounted for by the factors as follows: (i) The cross-linked structure formed in this reaction process of sulfur and pitch effectively prevented the melting of the pitch in the carbonization process, thus leading to a relative larger layer spacing for SPC-800. In addition, the cross-linked structure also provided a more stable structure. (ii) The increased layer spacing favored the rapid transfer of Li^+^/Na^+^. Moreover, diffusion-controlled intercalation process (DIP) behavior in SPC-800 was more significant, which resulted in a higher specific capacity and cyclic durability [44]. (iii) The presence of sulfur altered the inherent physicochemical characteristics of the samples (Figure 8); specifically, reducing the resistivity of SPC-800 (0.0268 Ω cm compared to 0.0381 Ω cm for PC-800) led to enhanced electrical conductivity and thus optimized electrochemical performance [13].

## 4. Conclusions

To summarize, an easy and effective method was adopted for the production of sulfur-modified pitch-based carbon. During the pre-treatment process, the residue rate of the pitch products increased from 39.84% to 88.13% with the introduction of sulfur due to the cross-linking reaction between sulfur and pitch at low temperatures. The carbon spacing expanded from 0.353 nm to 0.358 nm after the subsequent carbonization process. The cross-linked structures effectively prevented the pitch from melting and undergoing ordered reorganization in the carbonization stage. When employed as the anode in LIB, the sulfur-modified pitch-based carbon demonstrated an outstanding reversible specific capacity, reaching 478.1 mAh g^−1^. Similarly, a maximum reversible specific capacity of 220.9 mAh g^−1^ was achieved in SIBs. The superior behavior achieved is largely ascribed to the disordered structure, as well as to the presence of sulfur. This work highlights the immense potential of sulfur-modified pitch-based carbon materials for achieving exceptional battery performance and provides new perspectives on the regulation of carbon material microstructure through precursor design.

## Figures and Tables

**Figure 1 nanomaterials-14-01410-f001:**
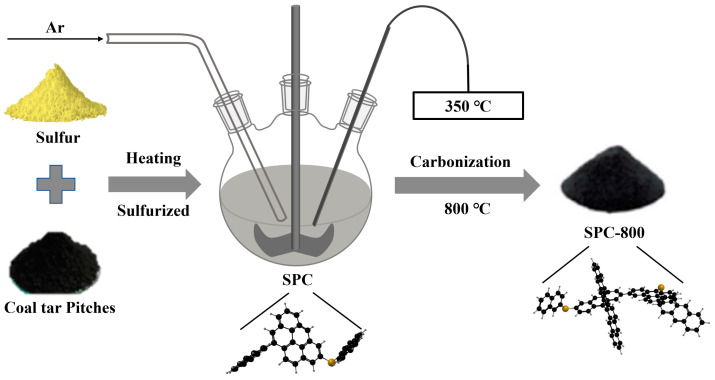
The process of preparing sulfurized pitch carbon.

**Figure 2 nanomaterials-14-01410-f002:**
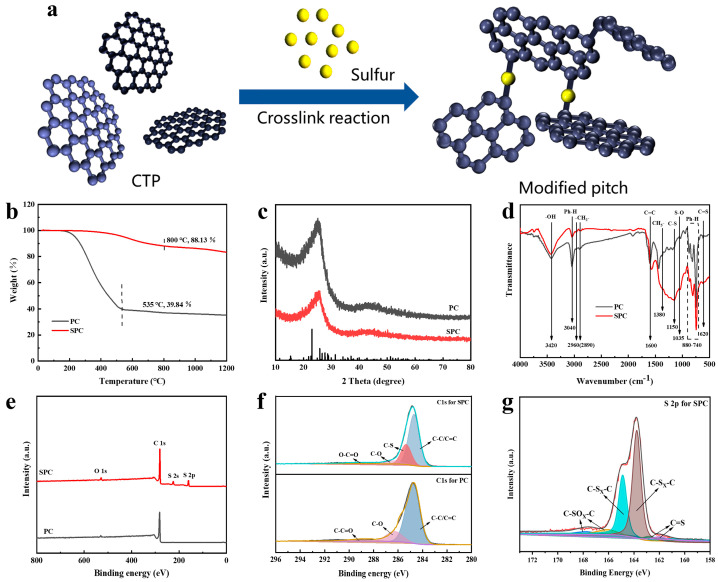
(**a**) Schematic diagrams of the synthesis of SPC, (**b**) TG curves, (**c**) XRD patterns, (**d**) FT-IR spectra, (**e**) XPS spectra, (**f**) C 1s spectra of PC and SPC, and (**g**) S 2p spectrum of SPC.

**Figure 3 nanomaterials-14-01410-f003:**
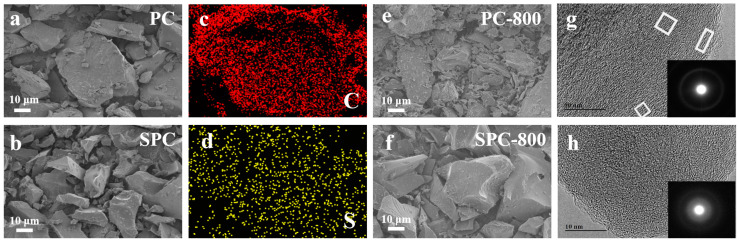
(**a**,**b**) SEM images of PC and SPC, (**c**,**d**) elements mapping of SPC, (**e**,**f**) SEM images of PC-800 and SPC-800, (**g**,**h**) HRTEM images of PC-800 and SPC-800.

**Figure 4 nanomaterials-14-01410-f004:**
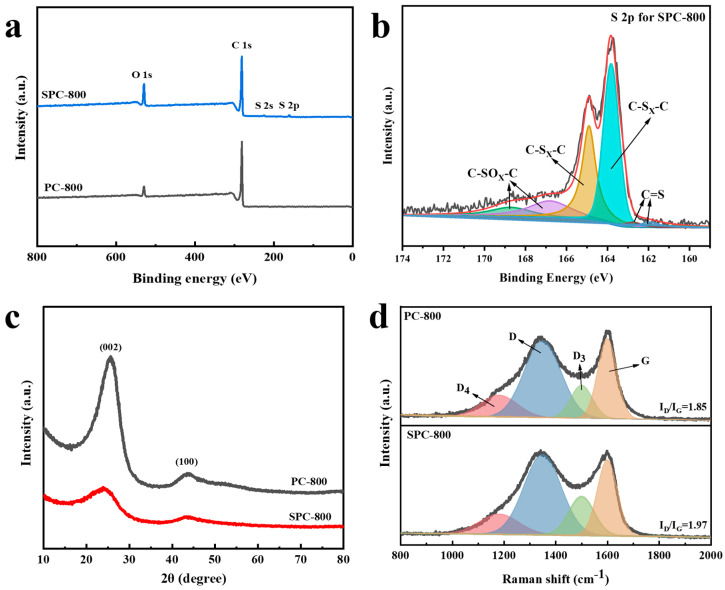
(**a**) XPS spectra of SPC-800 and PC-800, (**b**) S 2p spectrum of SPC-800, (**c**) XRD patterns of SPC-800 and PC-800, and (**d**) Raman spectra of SPC-800 and PC-800.

**Figure 5 nanomaterials-14-01410-f005:**
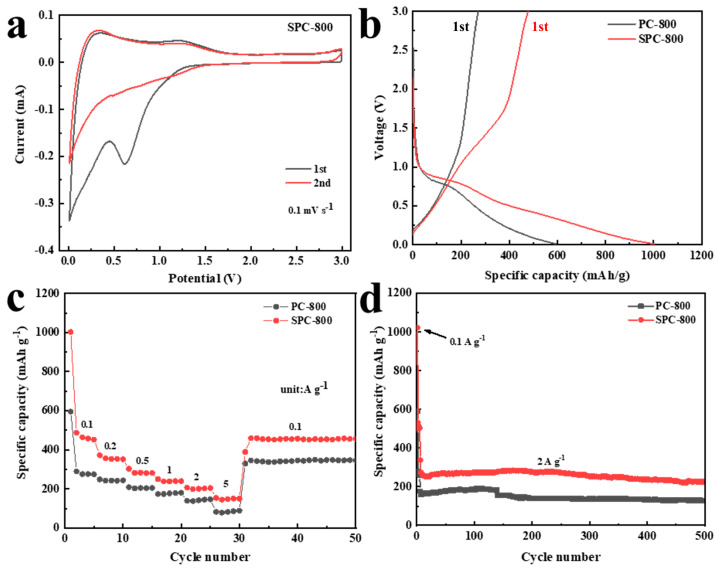
Electrochemical performance of PC-800 and SPC-800 as LIB anode materials: (**a**) cyclic voltammetry curves, (**b**) initial galvanostatic charge/discharge curves, (**c**) performance rate and (**d**) cyclic stability.

**Figure 6 nanomaterials-14-01410-f006:**
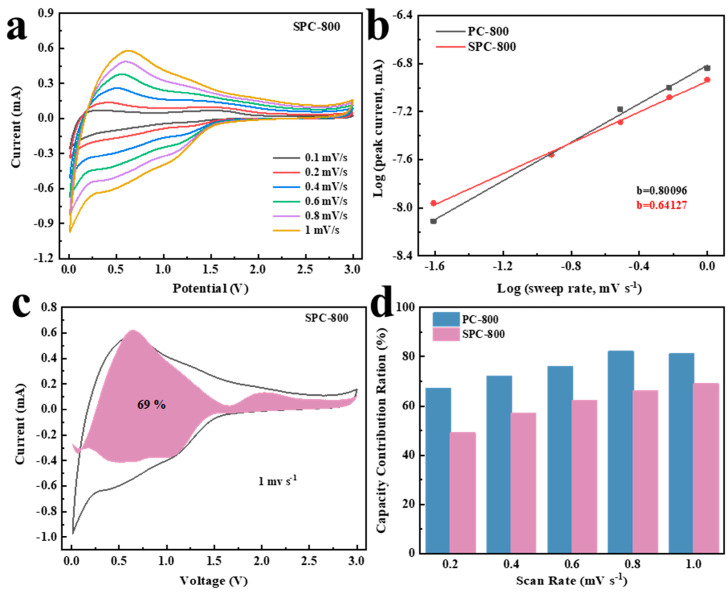
CV curves and capacitance contribution of PC-800 and SPC-800 as LIB anode materials: (**a**) Cyclic voltammetry curves at different scan rates, (**b**) Log (i) − log (v) plots, (**c**) capacitance contribution of SPC-800, and (**d**) capacitance contribution.

**Figure 7 nanomaterials-14-01410-f007:**
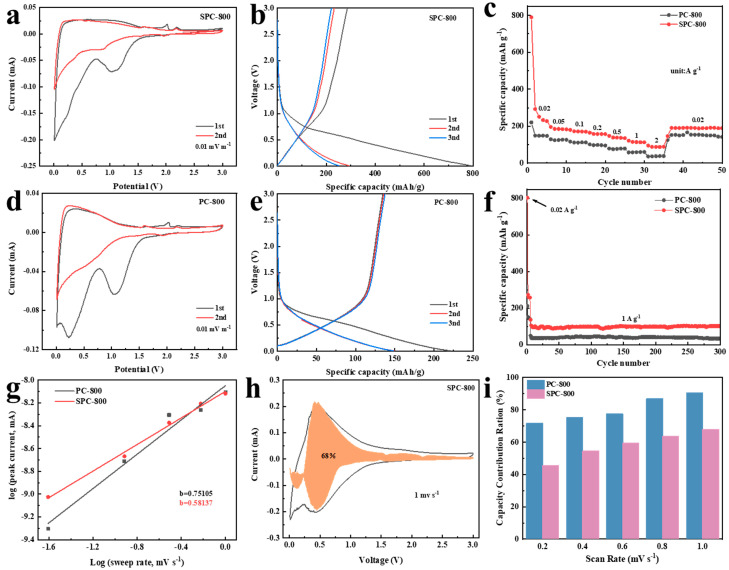
Electrochemical performance of PC-800 and SPC-800 as SIB anode materials: (**a**,**d**) cyclic voltammetry curves; (**b**,**e**) initial charge/discharge curves; (**c**) rate performance; (**f**) cyclic stability; (**g**) Log(*i*) − log (*v*) plots, (**h**) capacitance contribution of SPC-800, and (**i**) capacitance contributions.

**Figure 8 nanomaterials-14-01410-f008:**
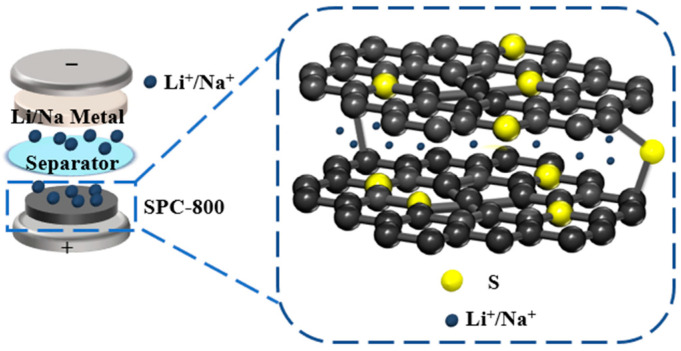
Relationship between the structure and electrochemical performance of SPC-800.

**Table 1 nanomaterials-14-01410-t001:** The characterization parameters of CTP.

C/%	H/%	O/%	N/%	S/%	SP/°C
93.17	4.38	0.63	0.95	0.64	110.5

**Table 2 nanomaterials-14-01410-t002:** Elemental analysis results of PC and SPC.

Samples	Elemental Composition (%)					C/H ^[b]^
	N	C	H	S	O ^[a]^	
PC	1.00	93.48	4.12	0.58	0.82	1.89
SPC	0.89	80.59	2.61	15.30	0.61	2.57

Note. ^[a]^ Calculated by subtraction method. ^[b]^ Molar ratio.

**Table 3 nanomaterials-14-01410-t003:** Comparison of electrochemical performance between SPC-800 and other anodes reported.

Samples	Capacity (mAh g^−1^)	Capacity Retention	Rate Capacity (mAh g^−1^)	Ref.
SPC-800	478.1 (0.1 A g^−1^)	100% (100 cycles, 5 A g^−1^)	150.5 (5 A g^−1^)	This work
CSC-0	311 (0.1 A g^−1^)	-	146 (5 A g^−1^)	[31]
AC@G	434.1 (0.1 C)	96.8% (100 cycles, 6 C)	115.3 (5 C)	[32]
NiP + G	460 (20 mA g^−1^)	-	220 (1 A g^−1^)	[33]
BCG	310 (0.1 A g^−1^)	95.3% (100 cycles, 0.2 C)	153.5 (5 A g^−1^)	[34]
BCNF	321 (0.1 C)	77% (50 cycles, 0.1 C)	50 (2 C)	[35]

Note: 1 C = 372 mA g^−1^.

## Data Availability

Data is contained within the article.

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
