# Peer review of "Effect of Sulfur Modification on Structural and Electrochemical Performance of Pitch-Based Carbon Materials for Lithium/Sodium Ion Batteries"

_nanomaterials, 2024, doi:10.3390/nano14171410_

Round 1

Reviewer 1 Report

Comments and Suggestions for Authors

In this manuscript, the authors reported a sulfur-modified carbon anode material for lithium-ion batteries and sodium-ion batteries. However, the results presented in the work was not fully studied. There are a number of key questions that need to be addressed before the paper can be accepted.

1. Abbreviations LIB, NIB, CTP are missing from the text.

2. One SIB must be used in the text and abstract to abbreviate sodium-ion batteries

3. What do the authors mean by the term "disordered nature"?

4. Why is capacity of SPC-800 (478.1 mAh g-1 at 0.1 A g-1) different from the capacity of sulfur-doped pitch-based carbon (482.8 m Ah g-1 at 0.1 A g-1) in Ref. 16.

5. In 2.1. Materials preparation: “For comparison, pitch carbon was prepared by treatment of CTP at 350 ℃ without the addition of sulfur. The corresponding carbonized product was labeled as PC-800.” Why was the temperature 350 °C and not 800 °C?

6. In Table 1: What does “SP/%” mean?

7. In Figure 1, the first stage “mixing” is called as heating and vulcanization in the experimental part. If heating was carried out with continuous stirring, this must be mentioned in the experimental part. The duration of the heating is missing in the experimental part. The name of this stage in the figure should be changed. The temperature for carbonization should be added in Figure 1.

8. 2.2. Materials Characterization, the name of the spectrometers must be written for FTIR and Raman spectroscopy. “Raman spectroscopy” must be used instead of “Raman”.

9. In 2.3. Electrochemical Characterization, LiPF6 is written incorrectly. What does PP mean? Why were different methods used to prepare active carbon electrode material for LIBs and SIBs?

10. What does “the aromatic C−H bond” mean? Hydrogen can not be included in the aromatic heterocycles because it has only one 1s valence orbital and no p orbitals.

11. In Fig. 2a: Why are carbon fragments bonded by only one S? Two or more sulfur chains can also form.

12. In Figure 2, there are two times “a”. TG profiles must be “b”. 

13. Analysis of TG profiles: “After heating treatment, the carbon residue of SPC was maintained up to 88.13% (that of PC was only 39.84%), demonstrating that the structure of SPC was more stable than PC”. For SPC, it is necessary to confirm that this is a carbon residue without sulfur.

14. The description of XPS survey and C 1s spectra are missing in the text.

15. Analysis of the XPS S 2p spectrum: “The high-resolution spectra of S 2p for SPC could be fitted into four distinct peaks, exhibiting binding energies of 162, 163.7, 164.8, and 168.4 eV, which respectively correspond to the C=S group, S2p3/2 and S2p1/2 of the thiophene sulfur (C−SX−C, x = 1, 2) covalent bond and S-oxide (C−SOX−C, x = 2, 4) [22, 23].” This is not correct. Due to the spin-orbit splitting, each sulfur state has 2p3/2 and 2p1/2 components, not only thiophene sulfur. Doublets, not single peaks, must use to fit the S 2p spectrum.

16. What about S-S bonds in the SPC and SPC-800? Why were they excluded from consideration?

17. The experimental results presented in the article are not sufficient to prove that major sulfur has formed bonds with carbon and there is no elemental sulfur in the sample. In SPC-8, the mass ratio of S to C is 8:20, which means the atomic ratio is 1:6. Due to the small number of S8 clusters, their reflexes may not be visible in the XRD. The XPS S 2p3/2 component of the spectrum of elemental sulfur is located around 164 eV and can make the main contribution to the spectrum of SPC. To prove the formation of S-C bonds, the XPS S 2p spectrum of SPC must be compared with the spectrum of sulfur (without the carbon component) after heating at 350 °C under the same conditions.

17. In the second reaction mechanism, it is necessary to clarify what “Sx” means.

18. It remains unclear how sulfur affected the graphitization of CTP? How did the size of the graphite stacks and graphite flakes change? These sizes can be estimated from width of XRD peaks.

19. “sp2−sp3” is written incorrectly.

20. The concentration of carbon, sulfur and oxygen in PC-800 and SPC-800 estimated from the XPS survey spectra should be compared. According to intensity of O 1s line, oxygen content in SPC-800 is higher than in PC-800. Why?

21. What is the difference between C 1s spectra of SPC-800 and PC-800?

22. In the XPS S 2p spectrum of SPC-800, C−SOx−C component must be fitted by a doublet.

23. Why does SPC-800 demonstrate two different capacity values 478.1 and 451.8 mAh g-1 at the same current density of 0.1 Ag-1?

24. In Fig.5 b, it is necessary to indicate the number of cycle.

25. Figures 5c and 5d are not mentioned in the text.

26. In the Table 3, “CSC-2, AG@G, NiP+G, BCG, BCNF” must be decrypted. The capacity of SPC-800 must be compared with the capacity of sulfur-containing carbon anode materials.

27. Captions for Figures 5, 6, 7 should retain information about whether the data refers to lithium or sodium. Potentials relative to lithium or sodium?

28. How do numerous oxygen-containing groups affect the structure and electrochemical properties of SPC-800?

Comments on the Quality of English Language

Although the language expression is clear, there still exist some minor grammatical, syntax or word usage errors in the manuscript. The English language should be carefully corrected.

Reviewer 2 Report

Comments and Suggestions for Authors

The manuscript is original and well written. I could not detect any criticism that may slow the publication. Therefore I recommend to accept it in  present form

Author Response

The manuscript is original and well written. I could not detect any criticism that may slow the publication. Therefore, I recommend to accept it in present form.

Response: Thanks very much for the positive comments.

Reviewer 3 Report

Comments and Suggestions for Authors

In the present paper, the authors have prepared sulfurized pitch-based carbons for use in lithium/sodium ion batteries. Despite the interesting topic, the paper should not be published in its present form.

Authors compare the electrochemical performance of SPC-800 (in Table 3) with electrodes based on graphite (C=372mA/g). Is the mechanism of reaction at the electrode the same?

In addition, the authors should use only one abbreviation: SIBs or NIBs for sodium-ion batteries. The authors should explain what type of parameters are presented in Table 1. There is an error in the description of Figure 2. Figure 5b does not show charge and recharge curves, it shows charge and discharge curves.

Author Response

Some content is not fully displayed, please see the attachment.

In the present paper, the authors have prepared sulfurized pitch-based carbons for use in lithium/sodium ion batteries. Despite the interesting topic, the paper should not be published in its present form.

Authors compare the electrochemical performance of SPC-800 (in Table 3) with electrodes based on graphite (C=372mA/g). Is the mechanism of reaction at the electrode the same?

Response: Thank you very much for your question. For the graphite electrode of LIBs, the mechanism is based on “insertion-filling” (diffusion-controlled behavior) mechanism, which is governed by diffusion. For SPC-800 in LIBs, there is more significant diffusion-controlled intercalation process in SPC-800. Therefore, the reaction mechanism of the electrode is the same in the diffusion process.

In addition, the authors should use only one abbreviation: SIBs or NIBs for sodium-ion batteries.

Response: We agree this comment. We uniform the abbreviation of sodium-ion batteries (SIBs) in the text.

The authors should explain what type of parameters are presented in Table 1.

Response: Thank you very much for the suggestion. We have refined the description in the revised manuscript.

Details are as follows:

CTP (Shanxi Yongdong Chemical Company, Characterization parameters, including elemental analysis and soft point outlined in Table 1) was employed as the raw material. It can be found in the revised manuscript-page 2, line 78-79.

There is an error in the description of Figure 2. Figure 5b does not show charge and recharge curves, it shows charge and discharge curves.

Response: Thank you very much for pointing this out. We have fixed the error in the description in Figure 2 and Figure 5b in the revised manuscript.

Details are as follows:

Fig. 5b exhibited the charge and discharge curves for both PC-800 and SPC-800. It can be found in the revised manuscript-page 7, line 251.

Round 2

Reviewer 1 Report

Comments and Suggestions for Authors

The paper may be considered for publication in the journal of Nanomaterials after major revisions. There are several unclear points in the manuscript that need to be clarified.

1.       Lines 87-88: The temperature of carbonization of PC-800 should be added in the text.

2.       Lines 97-99: For TG, XPS, FTIR, Raman scattering, and XPS, the names of the spectrometers and experimental parameters of measurements must be added.

3.       Line 111: Why different binders were used to prepare the electrode materials for LIB and SIB?. Why PVDF is not suitable for SIB anode?

4.       In Table 1, “SP” must be deciphered as “softening point” in the text.

5.       Lines 134-135: The statement “the carbon  residue of SPC was higher than PC” is not correct, since the mass residue of SPC consists not only of carbon, but also of sulfur.

6.       Line 140 and 159. Fig. 2d and Fig. 2f must be mentioned in the text.

7.       Lines 160-162, “It could be fitted into four distinct peaks, with binding energies of 162, 163.7, 164.8, and 168.4 eV, which correspond to the C=S group, S2p3/2 and S2p1/2 of the sulfur (C−SX−C, x = 1, 2) covalent bond and S-oxide (C−SOX−C, x = 2, 4), respectively [22, 23]”. This statement is incorrect. Each sulfur state (C=S, C-Sx-C, C-SOx-C) must be fitted by three doublets giving six (not four!) peaks.

Round 3

Reviewer 1 Report

Comments and Suggestions for Authors

The manuscript can be accepted after minor revisions.

In Fig.2g and Fig.4b, XPS S 2p spectra must to be refitted taking into account that area ratio for the two spin orbit peaks (2p1/2:2p3/2) must be 1:2. For each sulfur state, the width of 2p1/2 and 2p3/2 must be approximately the same.

Author Response

Some content is not fully displayed, please see the attachment.

  1. In Fig.2g and Fig.4b, XPS S 2p spectra must to be refitted taking into account that area ratio for the two spin orbit peaks (2p1/2:2p3/2) must be 1:2. For each sulfur state, the width of 2p1/2and 2p3/2 must be approximately the same.

Response: Thank you very much for your suggestion. We have refitted the peaks in the revised manuscript. Figure details are as follows:

It can be found in the revised manuscript-Fig. 2g and Fig. 4b.
